# Genome-wide association study identifies 14 previously unreported susceptibility loci for adolescent idiopathic scoliosis in Japanese

Ikuyo Kou et al.[#]

Adolescent idiopathic scoliosis (AIS) is the most common pediatric spinal deformity. Several AIS susceptibility loci have been identified; however, they could explain only a small proportion of AIS heritability. To identify additional AIS susceptibility loci, we conduct a meta-analysis of the three genome-wide association studies consisting of 79,211 Japanese individuals. We identify 20 loci significantly associated with AIS, including 14 previously not reported loci. These loci explain 4.6% of the phenotypic variance of AIS. We find 21 *cis*-expression quantitative trait loci-associated genes in seven of the fourteen loci. By a female meta-analysis, we identify additional three significant loci. We also find significant genetic correlations of AIS with body mass index and uric acid. The cell-type specificity analyses show the significant heritability enrichment for AIS in multiple cell-type groups, suggesting the heterogeneity of etiology and pathogenesis of AIS. Our findings provide insights into etiology and pathogenesis of AIS.

Correspondence and requests for materials should be addressed to C.T. (email: chikashi.terao@riken.jp) or to K.W. (email: watakota@gmail.com) or to S.I. (email: sikegawa@ims.u-tokyo.ac.jp). [#]A full list of authors and their affiliations appears at the end of the paper.

AIS is a complex spinal deformity which is defined as a lateral spinal curvature with a Cobb angle of >10 degrees[1]. AIS is a common disease affecting ~2.5% of adolescents in Japan[2] and millions of children worldwide are affected with a prevalence of 2–3%[3,4]. AIS is regarded as a multifactorial disease affected by genetic and environmental factor[5]. The importance of genetic factors in the etiology and pathogenesis of AIS has been demonstrated by many twin, family and population studies[6,7].

Genome-wide association study (GWAS) is one of the most effective methods to identify genetic factors of complex traits, including common diseases. To date, several GWASs have been conducted for AIS[8–13]. We previously conducted two GWASs (GWAS1 and GWAS2) in Japanese populations and identified three loci significantly associated with AIS susceptibility (10q24.31, 6q24.1 and 9p22.2)[8–10]. A subsequent meta-analysis using multi-ethnic cohorts confirmed their robust associations[14]. A total of seven significantly associated loci have been identified in the meta-analysis; however, the proportion of the AIS heritability explained by the seven loci is estimated to be only ~3%. Therefore, it is indispensable to identify additional susceptibility loci for understanding the etiology and pathogenesis of AIS.

In the present study, we perform a large Japanese GWAS followed by a meta-analysis of three GWASs. We replicate the association of 6 previous reported susceptibility loci and identify 14 previously unreported susceptibility loci. By a sex-stratified analysis, we further identify three significant loci for female AIS. The integrative analyses indicate the significant genetic correlations of AIS with body mass index (BMI) and uric acid (UA), and show the significant heritability enrichment for AIS in multiple cell-type groups. In addition, we find 21 cis-expression quantitative trait loci (eQTL)-associated genes in 7 out of 14 previously unreported susceptibility loci. In vitro functional analyses suggest that one of these eQTLs, rs1978060, regulates the expression of TBX1, and the difference in FOXA2 binding causes difference in cis-acting transcriptional regulatory function between alleles. These findings provide insights into etiology and pathogenesis of AIS.

## Results

**Association analysis**. To identify the additional AIS susceptibility loci, we conducted a large GWAS (GWAS3: 3254 cases and 63,252 controls). In addition, we reanalyzed the previous GWASs (GWAS1 and GWAS2) by updating the reference panel (Methods). For each of the three GWASs, imputation analysis was performed separately. Subsequently, the meta-analysis combining the three GWASs was performed (a total of 5327 cases and 73,884 controls; Supplementary Fig. 1). The genomic control inflation factor ($\lambda_{GC} = 1.16$) showed an inflation in the GWAS; however, the linkage disequilibrium (LD) score regression analysis indicated that the inflation was mostly from polygenicity (85.5%) and biases have a small contribution (estimated mean $\chi^2$ = 1.29 and LD score intercept = 1.04). Compared to the previously reported GWAS, the bias was not large[15]; we therefore did not apply the GC correction. As a result of the meta-analysis, we identified 20 significant AIS loci including 14 previously unreported loci (Table 1, Supplementary Data 1, Fig. 1 and Supplementary Fig. 2). The most significant association was identified at rs11190870 ($P = 2.01 \times 10^{-82}$) as with our previous GWASs[8,10]. Including this locus, significant associations of the six previously reported loci (10q24.31, 6q24.1, 9p22.2, 20p11.22, 16q23.3 and 9q34.2)[14] were replicated in the present meta-analysis (Table 1, Supplementary Data 1 and Fig. 1). Lead variants at the 14 previously unreported loci included both common genetic variants with smaller effect size and low-frequency genetic variants with relatively large effects sizes (Fig. 2 and Supplementary Table 1). Of the 14 variants, 3 variants had distinct minor allele frequency (MAF) spectra between Japanese (JPN) and Europeans (EUR). These three variants are rare (MAF < 0.01) or monomorphic in EUR, but low-frequency (MAF: 0.01–0.05) or common (MAF > 0.05) in JPN (Fig. 2 and Supplementary Table 1). We estimated the AIS heritability using LD score regression. The SNP heritability estimated on the liability scale by the LDSC software (https://github.com/bulik/ldsc) with the use of common variants in hapmap3 data is 42%. The lead variants of the 20 significant loci explained 4.6% of the phenotypic variance. We performed a stepwise conditional analysis to detect multiple independent signals within the 20 significant loci. We found five additional signals that reached the locus-wide significance ($P < 5.0 \times 10^{-6}$) (Supplementary Table 2). These five signals additionally explained 0.6% of the phenotypic variance (Supplementary Table 2 and Supplementary Fig. 3). These results were also confirmed by GCTA-COJO (Supplementary Table 3 and Supplementary Note 1).

**Integrative analysis**. AIS is a complex disease and several hypotheses including neuromuscular, biomechanical, genetic, developmental and growth-related have been proposed to explain the etiology and pathogenesis of AIS[5,16]. However, there is currently no hypothesis commonly supported. Therefore, the identification of AIS-related tissues and cell types is valuable for understanding of etiology and pathogenesis of AIS. We investigated the cell-type specificity of AIS based on the enrichment of heritability. We applied stratified LD score regression to our GWAS result using 220 cell-type-specific annotations of the four histone marks (H3K4me1, H3K4me3, H3K9ac and H3K27ac) constructed by the Roadmap Epigenomics Project[17]. First, to obtain an overview of AIS-related cell types, we assessed heritability enrichment of 10 major cell-type groups that are constructed reflecting system- or organ-level structures from the 220 individual cell-type-specific annotations[18]. The most significant enrichment was observed in the cardiovascular cell group ($P = 1.07 \times 10^{-5}$) (Supplementary Table 4). In addition, significant enrichment was also observed in other five cell-type groups: skeletal muscle, connective or bone, other, central nervous system (CNS) and gastrointestinal groups ($P < 5.0 \times 10^{-3}$) (Supplementary Fig. 4 and Supplementary Table 4). We further assessed the heritability enrichment of the 220 individual cell types and identified significant enrichments in six cell types ($P < 2.3 \times 10^{-4}$) (Supplementary Data 2). The most significant heritability enrichment was observed in H3K4me1 in fetal stomach cell ($P = 4.82 \times 10^{-6}$) and significant enrichment was also observed in H3K4me1 in other fetal organs such as lung, trunk muscle and leg muscle. Significant enrichment was also observed in H3K4me1 in stomach smooth muscle and H3K4me3 in penile foreskin fibroblast primary. These results suggest that the etiology and pathogenesis of AIS are heterogeneous and multifactorial.

To gain biological insights, we conducted a pathway analysis with the result of the meta-analysis using PASCAL[19] (https://www2.unil.ch/cbg/index.php?title=Pascal). Sixty-two pathways showed the nominal significance ($P < 0.05$); however, no specific pathway significantly associated with AIS ($P < 4.6 \times 10^{-5}$) was detected (Supplementary Data 3). To obtain insights into the genetic architecture of AIS, we explored the shared genetics between AIS and various traits. We calculated genetic correlation between this study and 67 complex human traits (61 quantitative traits and 6 diseases) in Japanese[20,21] using bivariate LD score regression[22] (Supplementary Data 4). Genetic correlations of AIS with BMI and UA have been suggested in our previous studies[20,21] and their significant negative correlations were replicated in this study (BMI: $r_g = -0.15$, $P = 3.14 \times 10^{-5}$; UA: $r_g = -0.15$, $P = 1.22 \times 10^{-5}$).

**Table 1 Association of the genome-wide significant loci**

| SNP | Chr. | Pos. | Gene in or near Region of association | RA | RAF Case | RAF Control | P value[a] | OR | 95% CI | $P_{het}$ | Genotyped/ imputation | Min Rsq |
|---|---|---|---|---|---|---|---|---|---|---|---|---|
| Previously unreported loci | | | | | | | | | | | | |
| rs141903557 | 4q21.23 | 85168056 | LOC101928978 | C | 0.060 | 0.047 | $9.78 \times 10^{-11}$ | 1.33 | 1.22-1.45 | 0.52 | Imputed | 0.98 |
| rs11205303 | 1q21.2 | 149906413 | MTMR11 | C | 0.24 | 0.21 | $1.62 \times 10^{-10}$ | 1.17 | 1.11-1.23 | 0.91 | Genotyped | 0.94 |
| rs12029076 | 1q42.13 | 228272687 | ARF1 | G | 0.81 | 0.78 | $2.17 \times 10^{-10}$ | 1.18 | 1.12-1.24 | 0.40 | Imputed | 0.99 |
| rs1978060 | 22q11.21 | 19749525 | TBX1 | G | 0.49 | 0.47 | $3.26 \times 10^{-10}$ | 1.16 | 1.11-1.22 | 0.62 | Imputed | 0.72 |
| rs2467146 | 12p12.3 | 17800607 | LINC02378/MIR3974 | A | 0.70 | 0.67 | $5.96 \times 10^{-10}$ | 1.15 | 1.10-1.20 | 0.23 | Imputed | 0.99 |
| rs11787412 | 8p23.2 | 3134239 | CSMD1 | A | 0.42 | 0.38 | $1.32 \times 10^{-9}$ | 1.14 | 1.09-1.18 | 0.86 | Imputed | 0.99 |
| rs188915802 | 9p13.3 | 34318683 | KIF24 | T | 0.019 | 0.013 | $1.94 \times 10^{-9}$ | 1.66 | 1.41-1.96 | 0.05 | Imputed | 0.81 |
| rs658839 | 6q14.1 | 81228722 | BCKDHB/FAM46A | G | 0.54 | 0.51 | $3.15 \times 10^{-9}$ | 1.14 | 1.09-1.19 | 0.88 | Imputed | 0.85 |
| rs160335 | 7p15.1 | 28587817 | CREB5 | G | 0.54 | 0.51 | $9.10 \times 10^{-9}$ | 1.13 | 1.08-1.18 | 0.69 | Imputed | 0.96 |
| rs482012 | 6q22.1 | 116430533 | NT5DC1 | T | 0.74 | 0.72 | $2.30 \times 10^{-8}$ | 1.14 | 1.09-1.19 | 0.15 | Genotyped | 0.99 |
| rs11341092 | 7p22.3 | 1269592 | LOC101927021/UNCX | AC | 0.33 | 0.31 | $2.92 \times 10^{-8}$ | 1.14 | 1.09-1.19 | 0.93 | Imputed | 0.81 |
| rs17011903 | 1q32.2 | 208259531 | PLXNA2 | A | 0.11 | 0.10 | $3.56 \times 10^{-8}$ | 1.20 | 1.13-1.28 | 0.11 | Imputed | 0.98 |
| rs397948882 | 7p21.2 | 15636869 | AGMO/MEOX2 | A | 0.11 | 0.10 | $3.66 \times 10^{-8}$ | 1.20 | 1.12-1.28 | 0.72 | Imputed | 0.98 |
| rs12149832 | 16q12.2 | 53842908 | FTO | G | 0.82 | 0.79 | $4.40 \times 10^{-8}$ | 1.16 | 1.10-1.22 | 0.34 | Genotyped | 0.96 |
| Previously reported loci | | | | | | | | | | | | |
| rs11190870 | 10q24.31 | 102979207 | LINC01514/LBX1 | T | 0.66 | 0.56 | $2.01 \times 10^{-82}$ | 1.52 | 1.46-1.59 | 0.22 | Genotyped | 1 |
| rs9389985 | 6q24.1 | 142653898 | ADGRG6 | G | 0.48 | 0.43 | $3.51 \times 10^{-20}$ | 1.21 | 1.16-1.26 | 0.43 | Imputed | 0.96 |
| rs7028900 | 9p22.2 | 16690612 | BNC2 | C | 0.46 | 0.42 | $2.19 \times 10^{-17}$ | 1.20 | 1.15-1.25 | 0.93 | Imputed | 0.93 |
| rs144131194 | 9q34.2 | 136145993 | ABO | AAGAAGGGAAA TTAATAAATATT | 0.58 | 0.55 | $1.35 \times 10^{-11}$ | 1.15 | 1.11-1.20 | 0.96 | Imputed | 1 |
| rs6047716 | 20p11.22 | 21894005 | PAX1/LINC01432 | C | 0.51 | 0.47 | $1.45 \times 10^{-11}$ | 1.15 | 1.11-1.20 | 0.77 | Imputed | 0.97 |
| rs2194285 | 16q23.3 | 82894817 | CDH13 | G | 0.13 | 0.11 | $8.69 \times 10^{-9}$ | 1.19 | 1.12-1.27 | 0.80 | Genotyped | 1 |

*Chr.* chromosome, *Pos.* genomic position (GRCh37/hg19), *RA* risk allele, *RAF* risk allele frequency, *OR* odds ratio, *CI* confidence interval, $P_{het}$ P values for heterogeneity from Cochran's Q-test
[a]The combined P values were calculated by the inverse-variance method under a fixed-effect model

**Sex-stratified analysis**. Since AIS has a clear sex difference of female predominance[23] and the number of male case samples was limited, we conducted a female-specific meta-analysis of the GWASs (5004 cases and 33,679 controls). As a result, 15 loci showed the genome-wide significance (Supplementary Data 5). Three of the fifteen loci (3q13.2, 1q25.2 and 1q23.3) did not reach GWAS significance in the overall study. We found an evidence of sex-heterogeneity in the *BOC* region on chromosome 3 (Table 2).

**Functional annotation and candidate susceptibility genes**. To identify candidate causal variants at each of the 14 previously unreported loci (Supplementary Fig. 2), we used several SNP annotation tools (HaploReg (https://pubs.broadinstitute.org/mammals/haploreg/haploreg.php), 3DSNP (http://cbportal.org/3dsnp/), RegulomeDB (http://regulomedb.org/), etc.) and explored the biological role of these variants. For each locus, we searched all variants in high LD ($r^2 > 0.8$ in East Asians (EAS) of 1KGP3) with the most associated variant (lead variant) and found some candidate variants with regulatory function (Supplementary Data 6). Furthermore, to identify susceptibility genes in the loci, we searched for eQTL using data from the Genotype-Tissue Expression (GTEx) project[24] (http://www.gtexportal.org/home/). As demonstrated by the cell-type-specific analysis, AIS is a multifactorial heterogeneous disease and multiple tissues may be associated with AIS. Moreover, because the tissues used in the GTEx project are limited and there are no data on AIS-related tissues such as intervertebral disc, cartilage and bone, we searched data on all tissues currently available. We observed significant *cis*-eQTLs at 7 of the 14 previously unreported loci, and the expression levels of 21 genes in the 7 *cis*-eQTL loci were associated with some variants that were in high LD ($r^2 > 0.8$ in EAS) with the lead variants (Supplementary Table 5). We also searched for candidate variants with regulatory functions and eQTL using GTEx data for three previously unreported loci that were significant in the female meta-analysis (Supplementary Data 7 and Supplementary Table 6). The expression levels of three genes in the chromosome 3 locus were associated with some variants that were in high LD ($r^2 > 0.8$ in EAS) with the lead variants (Supplementary Table 6). These *cis*-eQTL-associated genes are promising candidates for AIS susceptibility genes, among which there are several interesting candidate genes (Supplementary Table 7).

Among these *cis*-eQTLs, we specifically focused on rs1978060, the lead variant at chromosome 22q11.21 corresponding to the lead *cis*-eQTL variant of *TBX1*. *TBX1* is a member of the T-box gene family, which is a group of transcription factors involved in the regulation of developmental processes. Mutations of *TBX1* are known to cause DiGeorge syndrome (OMIM #188400) and velocardiofacial syndrome (OMIM #192430), and scoliosis is one of the clinical features, which is highly prevalent (47–49%) in association with these syndromes[25–28]. There is substantial evidence that *Tbx1* haploinsufficiency is responsible for the physical features of these syndromes, and it has also been reported that *Tbx1* knockout mice show vertebral anomalies[29,30]. In addition, other T-box gene family member, *TBX6* is known to cause congenital scoliosis[31]. We therefore selected the *TBX1* locus for further functional analysis. To prioritize candidate variants at the locus, we used a simple scoring system. The candidate variants were scored when they possess the following functional information: (i) promoter histone mark, (ii) enhancer histone mark, (iii) DNase protein binding and (iv) motif change. Based on the system, we selected rs1978060 as a most likely causal variant at this locus. rs1978060 was also a promising candidate causal SNP in other scoring systems such as 3DSNP and RegulomeDB (Supplementary Data 6).

We conducted in vitro analysis for rs1978060. We constructed luciferase reporter vectors by cloning the *TBX1* promoter region and inserting oligonucleotides that contained either the risk or non-risk allele of rs1978060. We evaluated the effect of rs1978060 on *TBX1* promoter activity and revealed that the risk allele-G significantly reduced the reporter activity compared to the non-risk allele-A (Fig. 3a and Supplementary Fig. 5a). We then performed an electrophoretic mobility shift assay to examine the DNA–protein binding of rs1978060. As expected, different binding patterns of DNA–protein complexes were observed between the risk and non-risk alleles of rs1978060 (Fig. 3b and Supplementary Fig. 5b). We searched for possible transcription factors that have the differential binding effect on rs1978060 using annotation tools. FOXA2 was predicted by JASPER (http://jaspar.genereg.net) to have a higher binding score in the non-risk allele and binding to this genomic region has also been demonstrated by chromatin immunoprecipitation assays. Furthermore, an electrophoretic mobility shift assay using FOXA2 antibody showed a super-shift in the presence of the antibody (Fig. 3c and Supplementary Fig. 5b) and the effect of

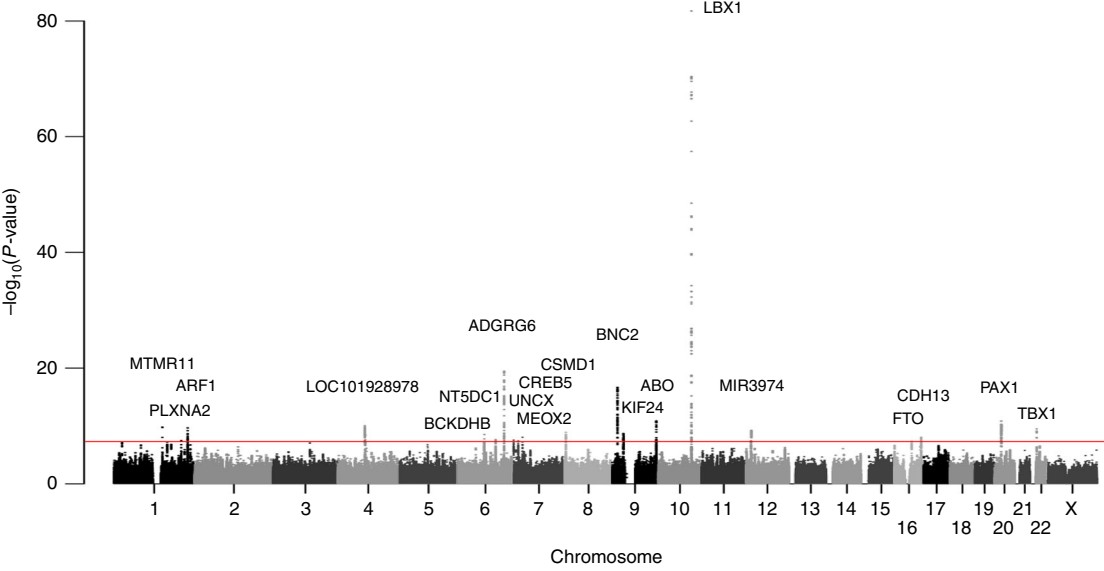

**Fig. 1** Manhattan plot showing the –log$_{10}$ P value for the SNPs in the AIS GWAS. P values were derived from the meta-analysis by using the inverse-variance method under a fixed-effect model. The red line represents the genome-wide significance threshold (P = 5 × 10$^{-8}$). The genetic loci that satisfied the genome-wide significance threshold in the meta-analysis of the three GWAS are labeled

rs1978060 on the *TBX1* promoter activity showed significant allelic difference when co-transfection with FOXA2 (Fig. 3a and Supplementary Fig. 5a). These results indicate that FOXA2 binds to rs1978060 and regulates the transcription of *TBX1*.

## Discussion

In the present study, we conducted a large AIS GWAS in Japanese and performed a meta-analysis with two previous Japanese GWASs. With larger sample sizes and improved imputation, we identified 20 susceptibility loci including 14 previously unreported loci. The larger sample size yielded both common variants with smaller effect sizes and low-frequency variants with relatively large effects on the risk of AIS, suggesting contribution of both rare and common variants in the genetic architecture of AIS (Fig. 2). Of the lead variants at the 14 previously unreported loci, 11 were common variants (MAF > 0.05) in both JPN and EUR. Some of these common variants differed greatly in MAF between JPN and EUR (Supplementary Table 1). However, the association of common lead variants (MAF > 0.05) at the three susceptibility loci (10q24.31, 6q24.1 and 9p22.2) identified in our previous GWAS was replicated even if there is a large difference in MAF between JPN and EUR[14,32,33]. Therefore, it is expected that the association of common lead variants (MAF > 0.05) identified in the present study will also be replicated in a multi-ethnic meta-analysis. From this point of view, rs73235136, a common lead variant (MAF > 0.05) at a female-specific susceptibility locus, also has a large difference in MAF between JPN and EUR, but the association is expected to be replicated. On the other hand, the low-frequency variants (rs141903557 and rs188915802), which have relatively large effects, are rare or monomorphic in EUR (Supplementary Table 1). The contribution of these variants with relatively large effects to the genetic architecture of AIS can differ among ethnic groups, as no association can be found unless there are other causal variants.

Besides our GWASs, a few GWASs in Caucasian and in Chinese have been reported[11–13]. The association of 20p11.22 locus reported in Caucasian GWAS was replicated in this study; however, the association of other loci reported in Chinese GWAS was not replicated. These loci were also not significant in our previous multi-ethnic meta-analysis[14]. They seem to be Chinese

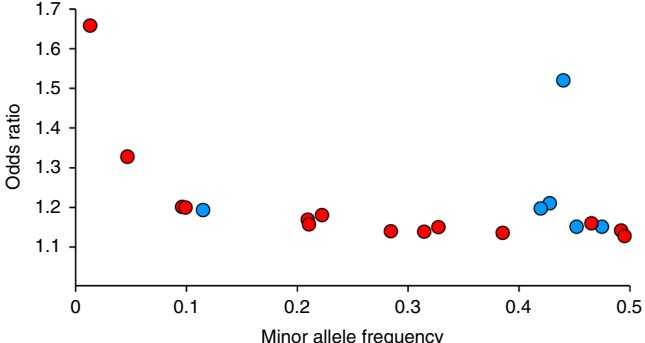

**Fig. 2** Relationship between effect size and minor allele frequency. The meta-analysis effect size (y axis) and the minor allele frequency (x axis) for 20 significant SNPs. Red circles represent the SNPs at previously unreported loci (n = 14) and blue circles represent the SNPs at previously reported loci (n = 6). Effect sizes are measured as odds ratios, which give the odds of the outcome given exposure to one risk allele compared with those to no risk allele

population-specific signals. Recently, an exome-wide association study identified that rs13107325, a non-synonymous SNP encoding SLC39A8, is associated with AIS[34]. However, the SNP is monomorphic in Japanese and other East Asian populations, and no variants associated with AIS at this locus were identified.

Integrative analyses indicated that AIS is associated with several cell-type groups such as cardiovascular, connective bone, skeletal muscle, and CNS, which suggests heterogeneity in etiology and pathogenesis of AIS. The negative correlation between AIS and BMI confirmed by our genetic analysis is consistent with the clinical observations reported in many clinical studies[35,36]. In contrast, there have been few reports on the relation between AIS and UA. We could find only a few papers reporting a relation between renal and ureteral abnormalities and congenital scoliosis[37,38]. UA is also known to be associated with a risk for cardiovascular diseases, and our cell-type specificity analysis showed that AIS is associated with cardiovascular cell types.

**Table 2 Association of the genome-wide significant loci for female AIS (only loci FEMALE-sig)**

| SNP | Chr. | Pos. | Gene in or near region of association | RA | F_RAF Case | F_RAF Control | F_OR (95% CI) | F_P value[a] | M_RAF Case | M_RAF Control | M_OR (95% CI) | M_P value[a] | $P_{het}$ |
|---|---|---|---|---|---|---|---|---|---|---|---|---|---|
| rs73235136 | 3q13.2 | 112951529 | BOC | C | 0.50 | 0.46 | 1.15 (1.10–1.20) | $3.45 \times 10^{-9}$ | 0.44 | 0.47 | 0.88 (0.75–1.03) | $1.01 \times 10^{-1}$ | 0.00057 |
| rs545608 | 1q25.2 | 177899121 | SEC16B | G | 0.76 | 0.73 | 1.16 (1.10–1.23) | $1.03 \times 10^{-8}$ | 0.75 | 0.74 | 1.05 (0.88–1.25) | $6.15 \times 10^{-1}$ | 0.13 |
| rs142502288 | 1q23.3 | 162450931 | SH2D1B/UHMK1 | G | 0.028 | 0.020 | 1.52 (1.31–1.76) | $3.11 \times 10^{-8}$ | 0.025 | 0.023 | 1.10 (0.66–1.84) | $7.13 \times 10^{-1}$ | 0.88 |

*Chr. chromosome, Pos. genomic position (GRCH37/hg19), F female, M male, RA risk allele, RAF risk allele frequency, OR odds ratio, CI confidence interval, $P_{het}$ P values for heterogeneity from Cochran's Q-test*
[a] The combined P values were calculated by the inverse-variance method under a fixed-effect model

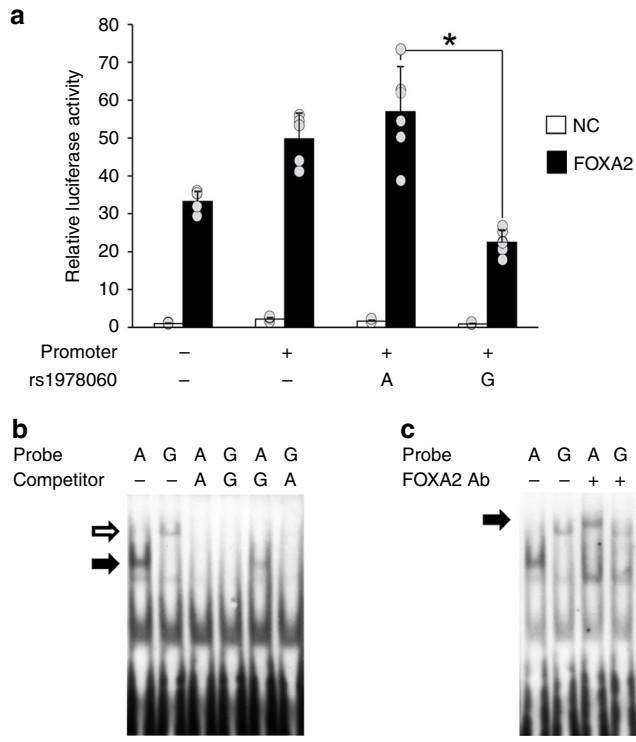

**Fig. 3** Allelic difference of functional variant, rs1978060 in Chr.22q11.21. **a** Reporter assays in MCF-7 cells. There was a significantly decreased transcriptional activity for the risk G-allele of rs1978060 compared to the non-risk A-allele. Error bars show standard deviation (S.D.) for each variant. Asterisks indicate statistically significant changes in paired comparison (*t*-test P < 0.01). n = 2 independent experiments. **b** Electrophoretic mobility shift assays with nuclear extracts from MCF-7 cells. There was specific bands for A-allele probe (lane 1, black arrow) and the G-allele probe (lane 2, white arrow) of rs1978060. Competition analyses were performed using an excess of the unlabeled A-allele probe (lane 3, lane 6) and G-allele probe (lane 4, lane 5) as competitors. **c** A super-shift assay using the FOXA2 antibody. The black arrow indicates a super-shifted FOXA2 complex in lane 3. Source data are provided as a Source Data file

These findings provide starting points to clarify the complex etiology and pathogenesis of AIS.

In the *cis*-eQTL analysis, we searched data for all tissues included in the current GTEx project; however, the tissues examined in the project are limited. AIS-related tissues such as intervertebral disc, cartilage and bone were not included and in the first place, it is difficult to determine the tissue that contributes to the onset of AIS. In addition, because the sample size greatly affects eQTL mapping, current eQTL data would have missed a considerable number of actually associated tissues. However, based on the *cis*-eQTL analysis followed by functional annotations of the variants, we could show that the *TBX1* expression is regulated through the binding of FOXA2 to rs1978060. Consistent with our results, previous mouse studies have shown that Foxa2 in the pharyngeal endoderm can bind and activate transcription through the critical *cis*-element upstream of *Tbx1* (refs. [39,40]).

Among the *cis*-eQTL-associated genes other than *TBX1*, there are several interesting candidate genes. One of them is *DSE* at chromosome 6q22.1. *DSE* encodes dermatan sulfate epimerase, an enzyme that is necessary for dermatan sulfate biosynthesis. Bi-allelic loss-of-function mutations in *DSE* are reported to cause Ehlers–Danlos syndrome (EDS) musculocontractural type 2 (OMIM #615539), which is characterized by progressive multi-

system fragility-related manifestations. The patients of this type of EDS present spinal deformity including scoliosis and kyphoscoliosis[36]. Consistent with this, *cis*-eQTL analysis shows that risk allele of rs482012 is associated with decreased expression of *DSE* (Supplementary Table 5). In addition, *FTO* on chromosome 16q12.2 is a promising candidate susceptibility gene. AIS-associated variants including the lead SNP, rs12149832 are present in the intronic region of *FTO* and the risk allele is associated with decreased expression of *FTO* (Supplementary Table 5 and Supplementary Data 6). It is well known that *FTO* is associated with BMI and obesity[41–43], and risk allele of rs12149832 is associated with lower BMI and decreased risk of obesity[43]. These findings are consistent with the recently reported negative correlation between BMI and AIS[20].

Sex-stratified analyses identified three susceptibility loci that are significant only in the female meta-analysis. Among the loci, we found an evidence of sex-heterogeneity at the *BOC* locus. *BOC* is a co-receptor for the hedgehog pathway which is induced by 1,25-dihydroxyvitamin $D_3$ (1,25(OH)$_2$D$_3$) in osteocytes[44]. Vitamin D is essential to maintain bone and mineral metabolism. Its status is greatly influenced by gender[45] and its deficiency is well known to be associated with osteoporosis and autoimmune disease which show significant sexual dimorphism[46]. Thus, *BOC* might be specifically associated with female AIS through the vitamin D metabolism.

In this study, 14 previously unreported AIS susceptibility loci were identified in Japanese, and *TBX1* was identified as one of the AIS susceptibility genes within the loci. Further studies are necessary to validate the association of these susceptibility loci in other ethnic groups. Identifying susceptibility SNPs within the loci and elucidating the function of candidate susceptibility would lead to understanding the etiology and pathogenesis of AIS. Although the mechanism behind sexual dimorphism in AIS remains unknown, the female-specific susceptibility locus we identified will be a key clue to understanding sexual dimorphism in AIS.

## Methods

**Subjects**. In GWAS3, as in GWAS1 and GWAS2, the case subjects with a Cobb angle of 10° or greater measured on standing spinal posteroanterior radiographs were recruited from collaborating hospitals[8–10]. All subjects were Japanese and were diagnosed with AIS between the ages of 10 and 18 years by expert scoliosis surgeons[8,47]. Congenital, juvenile, adult-onset scoliosis and scoliosis secondary to some other disorders were excluded. The control subjects were randomly selected from the subjects registered in the BioBank Japan project[48] (https://biobankjp.org/). For the quality control of GWAS samples, we removed samples with a call rate <0.98. We removed related individuals with PI_HAT > 0.25. PI_HAT is an index of relatedness between two individuals based on identity by descent implemented in PLINK[49] (https://www.cog-genomics.org/plink2). This filtering of relatedness was applied in each GWAS and combined data of the three GWAS separately and resulted in 102 cases and 806 controls excluded in total. To identify population stratification, we conducted principal component analysis (PCA) for genotype using FastPCA[50]. We excluded outliers from the East Asian cluster (distance from the mean of the cluster should be within 3 SD). There was no overlap of individuals in the three GWASs. Informed consents were obtained from all participants and from the parents of subjects who were minor. The ethical committees at all collaborating institutions and RIKEN approved the study.

**Genotyping and imputation**. GWAS3 subjects were genotyped by using the Illumina Human OmniExpress Genotyping BeadChip or a combination of Illumina HumanOmniExpress and HumanExome BeadChips. For quality control of variants, we applied the standard QC measures and excluded those with (i) SNP call rate < 0.99, (ii) MAF < 0.05 and (iii) Hardy–Weinberg equilibrium *P* value ≤ $1.0 \times 10^{-5}$. We also excluded the variants whose allele frequencies had differences of >0.16 between the GWAS dataset and the Asian data in reference panel. For each of the three GWASs, we pre-phased the genotypes using EAGLE and imputed dosages with the 1000 Genomes Project Phase 3 reference panel (May 2013 release; http://www.internationalgenome.org) with 1037 Japanese in-house reference panel using minimac3. For X chromosome, pre-phasing was performed in males and females together, and imputation was performed separately for males and females. Dosages of variants in X chromosomes for males were assigned between 0 and 2.

Since variants in the pseudo-autosomal region (PAR) were not contained in the reference panel, we did not analyze PAR. We used variants with an imputation quality score Rsq ≥ 0.3 and MAF ≥ 0.005 for the subsequent association study. The quality control and imputation analysis for three GWASs were processed separately. The number of SNPs through analysis steps is illustrated in Supplementary Table 8.

**GWASs and meta-analysis**. Association analyses of autosomes of GWAS1-3 were performed independently in a logistic regression model with top 10 principal components as covariates. The three GWAS were meta-analyzed with inverse-variance method under fixed effect model. For X chromosome, an association analysis was conducted separately for male and female at each GWAS, and meta-analyzed the association results. We filtered variants showing strong heterogeneity (Cochran's Q-test, *P* < 0.0001). This filtering excluded 484 variants in autosomes and 24 variants in X chromosomes. Regional association plots were produced by Locuszoom (http://locuszoom.org). Adjacent genome-wide significant (*P* < 5.0 × 10$^{-8}$) variants were grouped in one locus if they were located within 1 Mb apart from each other. To identify multiple independent signals within the 20 significant loci, we performed a stepwise conditional meta-analysis. We first performed conditional analyses of GWAS1-3 independently, and then combined the results using a fixed-effects model with the inverse-variance method. We repeated this process until the index variants fell below the locus-wide significance threshold of 5.0 × 10$^{-6}$, based on the approximate average number of multiple tests in each locus. Association studies were conducted by plink2 or mach2dat software. We estimated the AIS heritability using the LDSC software. The variance explained by SNPs was calculated based on a liability threshold model by assuming prevalence of AIS as 2.5%. In this model, we assume that subjects have a continuous risk score and that subjects whose score exceed a certain threshold develop AIS.

**Sex-specific GWAS**. We separately extracted imputed genotypes of male and female subjects from the GWAS data and applied the same quality control criteria in each GWAS of males and females. We empirically estimated difference in effect sizes in males and females. We randomly generated 100,000 true effect sizes in males and females based on correlation coefficients and standard errors in sex-specific GWAS and compared effect sizes between males and females to compute *P* values. We also calculated statistical power to identify GWAS significant signals in males and females (Methods).

**Power calculation of the current study**. We conducted power calculation with use of GeneticsDesign package of R software (https://www.r-project.org/) to detect a signal with *P* value of 5 × 10$^{-8}$ assuming disease prevalence of 3%. Calculation was done for entire dataset, males and females (Supplementary Table 9).

**Estimation of confounding biases using LD score regression**. To estimate confounding biases derived from population stratification and cryptic relatedness, we conducted LD score regression[15]. We used LD scores for the East Asian population provided by the LDSC software.

**Functional annotation and eQTL analyses**. To characterize associated variants, we used HaploReg v4.1, 3DSNP and RegulomeDB to gain functional annotation of variants in LD ($r^2$ ≥ 0.8 in EAS of 1KGP3) with the 14 AIS lead variants. LD was calculated using LDlink[51] (https://ldlink.nci.nih.gov), a web-based application. We searched for the overlaps between these associated variants and promoter and enhancer marks using HaploReg v4.1 setting the source for epigenetic annotation as ChromHMM core 15-state model. We also searched for the overlaps between the associated variants and lead *cis*-eQTL variants in GTEx (release v7). We considered only the *cis*-eQTLs with FDR < 0.05, and listed the variants showing the most significant association for each gene in Supplementary Tables 5 and 6.

**Cell-type specificity analysis**. To assess the heritability enrichment in cell types for AIS, we performed stratified LD score regression by combining data from specific annotations of 10 cell-type groups and 220 cell types and four activating histone marks (H3K4me1, H3K4me3, H3K9ac and H3K27ac) from the Roadmap Epigenomics project[17]. The variants with low imputation quality score (Rsq < 0.3) and the variants within the major histocompatibility complex (MHC) region (chromosome 6: 25–34 Mb) were excluded from the regression analysis. We defined significant heritability enrichments as those with *P* < 0.05 after Bonferroni correction.

**Pathway analysis**. To investigate biological pathways associated with AIS, we performed PASCAL[19]. PASCAL computes gene scores (max or sum scores) by aggregating SNP *P* values from a GWAS meta-analysis and calculates pathway scores by combining the scores of genes belonging to the same pathways. We used sum statistics and predefined pathway libraries from KEGG, REACTOME and BIOCARTA with default parameters.

**Genetic correlation analysis**. To estimate genetic correlations across the 67 traits (61 quantitative traits and 6 diseases), we conducted bivariate LD score regression[15] using the East Asian LD score and summary statistics of the current GWAS meta-analysis. We excluded SNPs found in the MHC region (chromosome 6: 25–34 Mb) from the analysis because of its complex LD structure. We defined significant genetic correlations as those with FDR < 0.05, calculated via the Benjamini–Hochberg method to correct multiple testing.

**Luciferase assay**. We constructed luciferase reporter vectors by cloning *TBX1* promoter (−912~ +63; TBX1-promoter-F: 5′-GTTGGTACCCTCCTCAGTGCT TCCCTTTG-3′ and TBX1-promoter-R: 5′-ACTCTCGAGAGTGTTCCTCCCTCC CTCAC-3′) with oligonucleotides (sense: 5′-TCGATGTCTAATGTACRCACCAG CTCGGA-3′and antisense: 5′-TCGATCCGAGCTGGTGYGTACATTAGACA-3′) containing either risk or non-risk alleles of rs1978060 into the multicloning site of a promoterless pGL4.10[luc2] vector (Promega). We also cloned a cDNA of FOXA2 (FOXA2-F: 5′-GTTAAGCTTGCCACCATGCACTCGGCTTCCAGTAT-3′ and FOXA2-R: 5′-ACTGGATCCAGAGGAGTTCATAATGGGCC-3′) into the multi-cloning site of p3xFLAG-CMV-14 vector (Sigma-Aldrich) for protein expression in mammalian cell lines. We sequenced the inserts of all constructs. The phRL-SV40 vector (Promega) was used as an internal control to normalize the variation in transfection efficiency. MCF-7 and OUMS-27 cells are generally used for luciferase assay. MCF-7 cells (HTB-22, ATCC) were cultured at 37 °C under 5% $CO_2$ in Dulbecco's modified Eagle's medium (DMEM) supplemented with 10% fetal bovine serum (FBS), 0.01 mg ml$^{-1}$ human recombinant insulin, 50 U ml$^{-1}$ penicillin and 50 μg ml$^{-1}$ streptomycin. OUMS-27 cells (JCRB Cell Bank) were cultured at 37 °C under 5% $CO_2$ in DMEM supplemented with 10% FBS and 50 μg ml$^{-1}$ kanamycin. All cell lines were authenticated by STR analysis and confirmed to be mycoplasma negative. MCF-7 and OUMS-27 cells were seeded in 24-well plates at a density of $5 \times 10^4$ and $1 \times 10^5$ cells per well, respectively. After 24 h, cells were transfected using TransIT-LT1 (Mirus Bio) according to the manufacturer's instructions. The luciferase activities were measured using the PicaGene Dual Sea Pansy Lumines-cence kit (Toyo Ink).

**Electrophoretic mobility shift assays**. We prepared probes for the risk (G) and non-risk (A) alleles of rs1978060 by annealing 25-bp complementary oligonu-cleotides (sense: 5′-TGTCTAATGTACRCACCAGCTCGGA-3′ and antisense: 5′-TCCGAGCTGGTGYGTACATTAGACA-3′) and labeling with digoxigenin (DIG)-11-ddUTP (Roche). The nuclear extracts were prepared from MCF-7 and FOXA2-overexpressing OUMS-27 cells. DNA–protein binding reactions were performed using a DIG gel shift kit according to the manufacturer's instructions (Roche). For competition assays, nuclear extracts were pre-incubated with excess unlabeled probes prior to adding DIG-labeled probes. For a super-shift assay, 2 μg of FOXA2 antibody (Santa Cruz; sc-374376X) was added to the reaction mixture and incubated for 20 min at room temperature. DNA-protein complexes were resolved on 6% DNA retardation gels (Thermo Fisher Scientific), and the signal was detected using a chemiluminescent detection system according to the manu-facturer's instructions (Roche). Uncropped gel images can be found in the Source Data file.

**Reporting summary**. Further information on research design is available in the Nature Research Reporting Summary linked to this article.

## Data availability

GWAS summary statistics of AIS are available at JENGER (Japanese ENcyclopedia of GEnetic associations by Riken, http://jenger.riken.jp/). Additional data used in this study are available from the corresponding authors upon reasonable request. The source data underlying Fig. 3a–c and Supplementary Figs. 5a and 5b are provided as a Source Data file.

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

## Acknowledgements

We thank all participating subjects and clinical staff at collaborating institutes. We also thank Y. Takahashi, T. Oguma, T. Kusadokoro, H. Takuwa and H. Suetsugu for technical assistance, and K. Sasada and Y. Yukawa for statistical assistance. This work was supported by JSPS KAKENHI Grant (no. 16H05453 to M.M., no. 18H02931 to I.K., no. 18H02932 to S.I.).

## Author contributions

S. Ikegawa designed the project and provided overall project management. I.K. and S. Ikegawa drafted the manuscript. I.K. performed functional analyses. Y.M. and M.K. performed the genotyping for the GWAS. C.T., H.F.L. and Y.K. analyzed the GWAS data and performed integrative analyses. C.T. and M. Nakajima contributed to preparing the manuscript. I.K., N.O., K.T., Y.O., Y. Takahashi, S.M., K.U., N.K., M.I., I.Y., Kei Watanabe, T. Kaito, H.Y., H.T., K.H., Y. Taniguchi, H. Shigematsu, T.I., S.D., R.S., N.F., M.Y., E.O., N.H., K. Kono, M.Nakamura, K.C., T. Kotani, T. Sakuma, T.A., T. Suzuki, K.N., K. Kakutani, T.T., H. Sudo, A.I., T. Sato, S. Inami, M.M. and Kota Watanabe collected and managed DNA samples and clinical data.

## Additional information

**Competing interests:** The authors declare no competing interests.

Ikuyo Kou[1,28], Nao Otomo[1,2,28], Kazuki Takeda[1,2], Yukihide Momozawa[3], Hsing-Fang Lu[1,4], Michiaki Kubo[3], Yoichiro Kamatani[5,6], Yoji Ogura[2], Yohei Takahashi[2], Masahiro Nakajima[1], Shohei Minami[7], Koki Uno[8], Noriaki Kawakami[9], Manabu Ito[10], Ikuho Yonezawa[11], Kei Watanabe[12], Takashi Kaito[13], Haruhisa Yanagida[14], Hiroshi Taneichi[15], Katsumi Harimaya[16], Yuki Taniguchi[17], Hideki Shigematsu[18], Takahiro Iida[19], Satoru Demura[20], Ryo Sugawara[21], Nobuyuki Fujita[2], Mitsuru Yagi[2], Eijiro Okada[2,22], Naobumi Hosogane[2,23], Katsuki Kono[2,24], Masaya Nakamura[2], Kazuhiro Chiba[2,23], Toshiaki Kotani[7], Tsuyoshi Sakuma[7], Tsutomu Akazawa[7], Teppei Suzuki[8], Kotaro Nishida[25], Kenichiro Kakutani[25], Taichi Tsuji[9], Hideki Sudo[26], Akira Iwata[27], Tatsuya Sato[11], Satoshi Inami[15], Morio Matsumoto[2], Chikashi Terao[5], Kota Watanabe[2] & Shiro Ikegawa[1]

[1]Laboratory for Bone and Joint Diseases, Center for Integrative Medical Sciences, RIKEN, Tokyo 108-8639, Japan. [2]Department of Orthopedic Surgery, Keio University School of Medicine, Tokyo 160-8582, Japan. [3]Laboratory for Genotyping Development, Center for Integrative Medical Sciences, RIKEN, Yokohama 230-0045, Japan. [4]School of Pharmacy, Taipei Medical University, Taipei 11042, Taiwan. [5]Laboratory for Statistical Analysis, Center for Integrative Medical Sciences, RIKEN, Yokohama 230-0045, Japan. [6]Kyoto-McGill International Collaborative School in Genomic Medicine, Kyoto University Graduate School of Medicine, Kyoto 606-8501, Japan. [7]Department of Orthopedic Surgery, Seirei Sakura Citizen Hospital, Sakura 285-8765, Japan. [8]Department of Orthopedic Surgery, National Hospital Organization, Kobe Medical Center, Kobe 654-0155, Japan. [9]Department of Orthopedic Surgery, Meijo Hospital, Nagoya 460-0001, Japan. [10]Department of Orthopedic Surgery, National Hospital Organization, Hokkaido Medical Center, Sapporo 063-0005, Japan. [11]Department of Orthopedic Surgery, Juntendo University School of Medicine, Tokyo 113-8421, Japan. [12]Department of Orthopedic Surgery, Niigata University Medical and Dental General Hospital, Niigata 951-8510, Japan. [13]Department of Orthopedic Surgery, Osaka University Graduate School of Medicine, Suita 565-0871, Japan. [14]Department of Orthopedic Surgery,

Fukuoka Children's Hospital, Fukuoka 813-0017, Japan. [15]Department of Orthopedic Surgery, Dokkyo Medical University School of Medicine, Mibu 321-0293, Japan. [16]Department of Orthopedic Surgery, Kyushu University Beppu Hospital, Beppu 874-0838, Japan. [17]Department of Orthopedic Surgery, Faculty of Medicine, The University of Tokyo, Tokyo 113-8655, Japan. [18]Department of Orthopedic Surgery, Nara Medical University, Kashihara 634-8522, Japan. [19]Department of Orthopedic Surgery, Dokkyo Medical University Koshigaya Hospital, Koshigaya 343-8555, Japan. [20]Department of Orthopedic Surgery, Kanazawa University Hospital, Kanazawa 920-8641, Japan. [21]Department of Orthopedic Surgery, Jichi Medical University, Shimotsuke 329-0468, Japan. [22]Department of Orthopedic Surgery, Saiseikai Central Hospital, Tokyo 108-0073, Japan. [23]Department of Orthopedic Surgery, National Defense Medical College, Tokorozawa 359-8513, Japan. [24]Department of Orthopedic Surgery, Eiju General Hospital, Tokyo 110-8645, Japan. [25]Department of Orthopedic Surgery, Kobe University Graduate School of Medicine, Kobe 650-0017, Japan. [26]Department of Advanced Medicine for Spine and Spinal Cord Disorders, Hokkaido University Graduate School of Medicine, Sapporo 060-8638, Japan. [27]Department of Preventive and Therapeutic Research for Metastatic Bone Tumor, Faculty of Medicine and Graduate School of Medicine, Hokkaido University, Sapporo 060-8638, Japan. [28]These authors contributed equally: Ikuyo Kou, Nao Otomo.

