## [Peer Review File · Nature Communications]

Reviewers' Comments:

Reviewer #1:

Remarks to the Author:

NCOMMS-19-03783-T

Kou et al. have performed a genome-wide association meta-analysis of AIS in Japanese individuals (n=79,211; 5,327 cases), identifying 20 loci of which 14 are claimed to be novel. Functional follow-up is done across the identified variants, including pleiotropy analysis (in the form of genetic correlations and tissue expression enrichment). Specific functional assessment is done on TBX1 one of the novel loci identified by the effort. The paper is well written and the methodology is sound.

Below some aspects (in no particular order) needed to be addressed by the authors:

1. It would be informative to see broader discussion and comparison between the findings in Europeans and Asians. Are there major allele frequency in the associated alleles? Is it expected that cross-ethnic meta-analysis replicate these findings? Can lookups of the effect sizes and other summary statistics be done in the European GWAS?

2. Please add a column with cytoband to Table 1 in order to follow the descriptions along the text using cytoband nomenclature. Similarly, please split Table 1 into loci where novel associations are reported and those arising from previous studies (ADGRG6, BNC2, ABO, LINC01514/LBX1, CDH13 and PAX1/LINC01432).

3. On top of or rather than the typical Manhattan Plot it would be more informative if a figure of MAF vs effect size is plotted for the top SNP's with different legend for known and novel loci. This will reinforce the discussion of genetic architecture to see if the increase in sample size is yielding more common variants or if there is a trend towards a lower minor allele frequency. Eyeballing through Table 1 it seems there may be a combination of both.

4. Related to the previous point please discuss and provide a power calculation to place in context of the identification of variants with MAF less than 5% or even < 1%.

5. Please elaborate further on the potential sex-specific associations and give more detail on the methodology (i.e., software and statistical test used, coding of hemizygous calls, management of pseudo-autosomal regions). This reviewer finds the sex-specificity claim for associations outside the sex chromosomes rather dodgy. Again, place this in the context of power scenarios (across sexes). Instead of showing all meta-analytical phases in Table 2 please show the effect estimates in men and women providing specific tests of heterogeneity of the estimates. Are they truly significantly different or this is just consequence of random variation in less powered strata?

6. Please provide more information on the variants with high heterogeneity that were excluded from the analysis, similarly provide number of subjects excluded from the analysis (IBD/IBS outliers).

7. There are more comprehensive reliable methods for conditional test for signal independence (allelic heterogeneity) like those implemented on GCTA (COJO). I strongly suggest to use them rather than the employed empirical method used by the authors.

8. The procedures used to assess functional annotations of ENCODE elements (using HaploReg, RegulomeDB, etc) and eQTL characterization are ill defined. Please discuss the choice of cell types and tissues and how are this relevant to the identified associations.

9. The current structure of the paper makes it difficult to read. While the rationale behind

prioritizing TBX1 for focused functional workup is presented, this nicely done experiment falls in the middle of the paper, removing attention from all subsequent functional work of broader scope. I suggest presenting the TBX1 results last which in many ways integrates the findings along the different segments of the paper.

10. Similarly, one of the reasons to prioritize TBX1 is that mutations cause DiGeorge and velocardiofacial syndrome, where subsequently it is stated that these conditions have scoliosis as important clinical manifestations. This is highly inaccurate as to my knowledge scoliosis is not a hallmark of any of these conditions specifically; rather, there is one report studying in aggregate the del22q11 syndrome (region comprising more than 20 genes) pointing out that scoliosis is a less frequently appreciated feature. Therefore, attention should be provided to the region 22q11.2 as harboring mutations that are associated with scoliosis presentation but not necessarily to these conditions.

11. Recently a non-synonymous SNP (rs13107325) encoding SLC39A8 (ZIP8 transporter involved in magnesium metabolism) was reported in this journal to be associated with AIS. Was there any evidence for association for these variant or any other in that locus?

Fernando Rivadeneira

Reviewer #2:

Remarks to the Author:

Kou et al have performed a meta-analysis of three GWAS on adolescent idiopathic scoliosis (AIS) in Japanese population, a common disease with a strong genetics component, which causes a complex spinal curvature. Although several GWASs have already been performed in different populations for this trait, identifying a number of genes, only 3% of the heritability of the trait has been explained. It is, therefore, necessary to carry out further studies to improve the knowledge of the genetics architecture of the disease. The current study on 5,327 cases and 73,884 controls of Japanese origin validated some previous findings and identified 14 novel loci. Altogether, these results explained 4.6% of the phenotypic variance. The authors also carried out several approaches to identify cell types involved, finding that the AIS is a very complex disease affecting multiple tissues, including cardiovascular tissue, bone, and skeletal muscle. Furthermore, Kou et al selected one of the novel targets in chromosome 22 for some functional analysis to identify the causal gene in this locus.

The genetics analysis undertaken in this study is rigorous and of high quality. They have used advanced genetic techniques to identify novel markers for AIS and also performed functional studies to validate one of them. However, the following concerns have arisen during the review process:

- The authors do not clarify the number of patients and controls recruited for the GWAS3, neither summarise the figures for the previous GWAS, for an easier understanding of the analysis. It seems that the greatest association for the novel markers is driven by the GWAS3, therefore it is important to describe in detail the recruitment process in comparison with the previous GWASs to understand how these novel signals were detected.

- Among the 14 novel loci, the authors selected rs1978060 on chromosome 22 for further functional analysis. However, they do not explain the reasons for the selection of the SNP, but not any of the other signals. It is understandable that previous work on TBX1 gene supports it as a candidate gene for AIS, but so it does the work on DSE (and FTO). As the authors well discuss, DSE causes Ehlers-Danlos syndrome, which involves scoliosis, similarly to TBX1 causing DiGeorge syndrome and velocardiofacial syndrome. Therefore, some functional work on DSE gene would have been expected.

- Regarding the functional work performed on TBX1, two big issues arise: the model selected and the methodology. It is well accepted that cell lines are not the most reliable model to investigate gene functionality, and if selected, the results should be validated in other cell lines, primary

cultures, or zebrafish/mouse models. In the present study only MCF7, a breast adenocarcinoma cell line, was used. The relevance of this cell line to mimic AIS features would be very limited if any. The lack of validation in other model reduces the relevance of the findings. The other issue arisen is the methodology. A very simplistic approach was used in order to provide functional information on this locus. The SNP selected is also a strong eQTL in blood for GNBIL gene, located next to TBX1 and also associated with the velocardiofacial syndrome. The lack of further investigation on the TBX1 gene and its functional role in the development of AIS raises the doubt if GNBIL and not TBX1 would be the causal gene, or if both genes could work complementary on the studied trait. With this very little work on the TBX1, and no mechanism proposed, these results are of limited value.

In order to accomplish the high standards of a journal like Nature Communications, I would suggest the authors to address the abovementioned concerns prior publication.

Moreover, some minor comments are suggested to improve the quality of the paper:

- It would be advantageous if the authors make the genetics data available for the scientific community who would like to replicate this work.
- Regarding the GWAS QC, cryptic relationships should also be taken into account when performing the general QC on each GWAS, not only at the final step as done in this study.
- It is unclear how the authors selected the causal SNP in each locus after the functional annotation. A ranking method should have been used to identify the SNP with highest functional qualities to be proposed as the one driving the association. Nowadays, there are various ranking software, like SuRFR, that can be used for this purpose.
- The authors only discussed the biological implication of three out of 14 novel loci identified in the study. It would be important to include a summary of the known biological role for the remaining loci.
- When performing principal component analysis, the authors did not clarify the threshold used to exclude individuals.
- In the introduction, the prevalence for AIS is referred to the USA population, but no information is provided regarding the Japanese population.
- The authors miss to review one of the latest publications on AIS using exome sequencing, published in Nature Communications in 2018 (Haller et al).
- The minor allele frequency threshold for imputed SNPs (0.005) is too small. It might be a typo, otherwise, the statistical power in this study would not be enough to identify reliable signals at this very low frequency.
- Table 1 needs to include minor allele frequency data.
- The number of SNPs genotyped and imputed used for association analysis needs to be listed. It would also be good to have information whether the SNPs listed in Table 1 were genotyped or imputed.

Answer to Reviewers' comments:

to Reviewer #1

1. It would be informative to see broader discussion and comparison between the findings in Europeans and Asians. Are there major allele frequency in the associated alleles? Is it expected that cross-ethnic meta-analysis replicate these findings? Can lookups of the effect sizes and other summary statistics be done in the European GWAS?

As suggested, we have added discussion and comparison between the findings in Europeans and Asians (Page 8, lines 138-141 and Pages 14-15, lines 264-277). Yes, most of associated alleles in novel loci are common variants. Based on our previous experiences [Ogura *et al.* Sci Rep 2018, Kou *et al.* Sci Rep 2018, Takeda *et al.* J Hum Genet 2019], it is expected that cross-ethnic meta-analysis will replicate these findings. No, lookups of the effect sizes and other summary statistics cannot be done in the European GWAS. There is no GWAS with comparative scale with this study.

2. Please add a column with cytoband to Table 1 in order to follow the descriptions along the text using cytoband nomenclature. Similarly, please split Table 1 into loci where novel associations are reported and those arising from previous studies (ADGRG6, BNC2, ABO, LINC01514/LBX1, CDH13 and PAX1/LINC01432).

As suggested, we have added a column with cytoband to Table 1 in order to follow the descriptions along the text using cytoband nomenclature. Also, we have split Table 1 into loci where novel associations are reported and those arising from previous studies.

3. On top of or rather than the typical Manhattan Plot it would be more informative if a

figure of MAF vs effect size is plotted for the top SNP's with different legend for known and novel loci. This will reinforce the discussion of genetic architecture to see if the increase in sample size is yielding more common variants or if there is a trend towards a lower minor allele frequency. Eyeballing through Table 1 it seems there may be a combination of both.

As suggested, we have added a figure of MAF vs effect size with different legends for known and novel loci (Supplementary Fig. 3). We have also discussed the genetic architecture of AIS from the trends seen by increasing the sample size (Page 14, lines 259-264).

4. Related to the previous point please discuss and provide a power calculation to place in context of the identification of variants with MAF less than 5% or even $< 1\%$.

As suggested, we have added a Supplementary Table of power calculation (Supplementary Table 14) and discussed a power of the current study to identify variants with low allele frequencies in Supplementary Note.

5. Please elaborate further on the potential sex-specific associations and give more detail on the methodology (i.e., software and statistical test used, coding of hemizygous calls, management of pseudo-autosomal regions). This reviewer finds the sex-specificity claim for associations outside the sex chromosomes rather dodgy. Again, place this in the context of power scenarios (across sexes). Instead of showing all meta-analytical phases in Table 2 please show the effect estimates in men and women providing specific tests of heterogeneity of the estimates. Are they truly significantly different or this is just consequence of random variation in less powered strata?

As suggested, we have elaborated further on the potential sex-specific associations and give more detail on the methodology. We have placed it in the context of power scenarios (across sexes).

Regarding the sex-specific associations outside the sex chromosomes, many studies have suggested that variations within the autosomal genomes of many species affect anatomical, physiological, and behavioral traits differently in males and females. In fact, thousands of genes with sex-biased expression are found on the autosomes [Ober *et al.* Nat Rev Genet. 2008]. In addition, the effects of autosomal genetic variants could be influenced by the products of genes not on sex chromosomes (ex. hormones), which can lead to sex specificity. Thus, differences in the incidence of diseases between the sexes do not necessarily mean that the disease gene is on sex chromosomes.

Thank you very much for the valuable comment on the effect estimate. Following the reviewer's comment, we have shown the effect estimates in men and women providing specific tests of heterogeneity of the estimates as follows.

First, we comprehensively re-structured the description of female-specific associations in methods. We made it clear that we extracted female/male imputed dosages from the entire GWAS and did NOT specifically run imputation for sex-specific associations. Thus, software and tests are the same as the entire GWAS. We also modified the parts of methods for overall data sets to include PAR. We excluded the PAR region in the current study. We added a sentence of dosages of variants in X chromosome in males (Page 20, lines 366-368).

Second, we carefully re-analyzed the female-specific associations. We found additional sex information which was not incorporated in the previous draft. In the revised manuscript, we used the updated sex information. As a result, two additional loci reached GWAS significant level in female GWAS which did not reach GWAS significant level in overall associations.

Third, we have entirely modified Table 2 to show the three variants exceeding the GWAS significant level in the female AIS GWAS, but not in the overall GWAS. We put statistic information in the male AIS GWAS and added empirical p-values to assess heterogeneity. We randomly generated 100,000 ‘true’ effect sizes in females and males based on correlation coefficients and standard errors in the sex-specific GWAS and compared effect sizes between females and males to compute p-values. As a result, we found an evidence of sex-heterogeneity in one locus. The previous Table 2 was moved to Supplementary Table 7.

Forth, we added tables of statistical power for males and females (Supplementary Table 14).

6. Please provide more information on the variants with high heterogeneity that were excluded from the analysis, similarly provide number of subjects excluded from the analysis (IBD/IBS outliers).

As suggested, we have provided information on the variants with high heterogeneity that were excluded from the analysis and number of subjects excluded from the analysis as follows. We excluded 484 variants in autosomes and 24 variants in X chromosomes due to high heterogeneity across the studies (Cochran’s $Q < 0.0001$) (Page 21, lines 379-380). We excluded 102 cases and 806 controls from the analyses due to kinship (Page 19, lines 342-346).

7. There are more comprehensive reliable methods for conditional test for signal independence (allelic heterogeneity) like those implemented on GCTA (COJO). I strongly suggest to use them rather than the employed empirical method used by the authors.

Thank you for the comments. In the revised manuscript, we have added the results using GCTA (COJO) as Supplementary Table 2 and Supplementary Note. As expected, the results of the two analyses (traditional conditional analysis and GCTA-COJO) were quite similar to each other. Since GCTA COJO took LD structure of reference data and generalize it to obtain statistics, we believe that results of the traditional conditional analysis (with the use of individual genotypes and imputed dosages) are worth reported at first.

8. The procedures used to assess functional annotations of ENCODE elements (using HaploReg, RegulomeDB, etc) and eQTL characterization are ill defined. Please discuss the choice of cell types and tissues and how are this relevant to the identified associations.

As suggested, we have presented the procedures used to assess functional annotations of ENCODE elements and eQTL characterization (Pages 11-12, lines 205-219 and Page 13, lines 231-237). Because the publicly available eQTL databases do not contain data on AIS related tissues, we searched data on all tissues currently available. We have discussed the choice of cell types and tissues in the text (Page 16, lines 289-294).

9. The current structure of the paper makes it difficult to read. While the rationale behind prioritizing TBX1 for focused functional workup is presented, this nicely done experiment falls in the middle of the paper, removing attention from all subsequent functional work of broader scope. I suggest presenting the TBX1 results last which in many ways integrates the findings along the different segments of the paper.

According to the suggestion, we have presented the TBX1 results last.

10. Similarly, one of the reasons to prioritize TBX1 is that mutations cause DiGeorge and velocardiofacial syndrome, where subsequently it is stated that these conditions have scoliosis as important clinical manifestations. This is highly inaccurate as to my knowledge scoliosis is not a hallmark of any of these conditions specifically; rather, there is one report studying in aggregate the del22q11 syndrome (region comprising more than 20 genes) pointing out that scoliosis is a less frequently appreciated feature. Therefore, attention should be provided to the region 22q11.2 as harboring mutations that are associated with scoliosis presentation but not necessarily to these conditions.

There are some reports showing that scoliosis is highly prevalent (47-49%) in association with del22q11 syndrome [Homans *et al.* Arch Dis Child 2019, Bassett *et al.* Am J Med Genet A. 2005]. However, to prevent misunderstanding of the relationship between scoliosis and del22q11 syndrome, we have rewritten the sentence (Pages 12-13, lines 223-226). After all, it was just one of the reasons to prioritize *TBX1*. Anyway, we could show through subsequent experiments that *TBX1* is the very likely susceptibility gene in the locus.

11. Recently a non-synonymous SNP (rs13107325) encoding SLC39A8 (ZIP8 transporter involved in magnesium metabolism) was reported in this journal to be associated with AIS. Was there any evidence for association for these variant or any other in that locus?

No, there was no evidence for association for these variants or any other in that locus. The SNP (rs13107325) is monomorphic in Japanese. The results were

mentioned in the text (Page 15, lines 274-277).

to Reviewer #2

- The authors do not clarify the number of patients and controls recruited for the GWAS3, neither summarise the figures for the previous GWAS, for an easier understanding of the analysis. It seems that the greatest association for the novel markers is driven by the GWAS3, therefore it is important to describe in detail the recruitment process in comparison with the previous GWASs to understand how these novel signals were detected.

We have clarified the number of patients and controls recruited for the GWAS3, and summarized the figures for the previous GWAS (Page 7, line 121 and Supplementary Fig. 1). The recruitment process of GWAS3 is included in the revision (Page 19, lines 335-341).

- Among the 14 novel loci, the authors selected rs1978060 on chromosome 22 for further functional analysis. However, they do not explain the reasons for the selection of the SNP, but not any of the other signals. It is understandable that previous work on TBX1 gene supports it as a candidate gene for AIS, but so it does the work on DSE (and FTO). As the authors well discuss, DSE causes Ehlers-Danlos syndrome, which involves scoliosis, similarly to TBX1 causing DiGeorge syndrome and velocardiofacial syndrome. Therefore, some functional work on DSE gene would have been expected.

We have explained the reasons for the selection of the SNP (Pages 12-13, lines 209-237). *DSE* and *FTO* are also very good candidate genes as we described

and the reviewer commented. We are working on functional experiments on the genes. We have not obtained conclusive data yet.

- Regarding the functional work performed on *TBX1*, two big issues arise: the model selected and the methodology. It is well accepted that cell lines are not the most reliable model to investigate gene functionality, and if selected, the results should be validated in other cell lines, primary cultures, or zebrafish/mouse models. In the present study only MCF7, a breast adenocarcinoma cell line, was used. The relevance of this cell line to mimic AIS features would be very limited if any. The lack of validation in other model reduces the relevance of the findings.

As suggested, the results in MCF-7 cells are validated in other cell line, OUMS-27 (a human chondrosarcoma cell line) (Supplementary Fig. 6).

- The other issue arisen is the methodology. A very simplistic approach was used in order to provide functional information on this locus. The SNP selected is also a strong eQTL in blood for *GNBIL* gene, located next to *TBX1* and also associated with the velocardiofacial syndrome. The lack of further investigation on the *TBX1* gene and its functional role in the development of AIS raises the doubt if *GNBIL* and not *TBX1* would be the causal gene, or if both genes could work complementary on the studied trait.

As the reviewer pointed out, the selected SNPs have strong eQTL in the *GNBIL* gene in blood. *GNBIL* is mapped within critical region of DiGeorge syndrome on chromosome 22q11.2 and may contribute to the etiology of those diseases. However, there is substantial evidence that haploinsufficiency for *TBX1* plays a role in the physical features of del22q11.2. In addition, *Tbx1* knockout mice show vertebral anomalies. In contrast to *TBX1*, there is no *GNBIL* mutation that has been reported to cause the del22q11.2 syndrome. *GNBIL* is known to be

associated with schizophrenia and autism [Williams *et al.* Hum Mol Genet. 2008; Chen *et al.* Am J Med Genet B. 2012]. Thus, *GNBIL* may be responsible for the spectral behavioral phenotype observed in del22q11.2 syndrome. However, there is no convincing evidence that *GNBIL* causes the del22q11.2 syndrome, especially its physical features. For these reasons we selected the *TBX1* for further investigation. We have added these reasons for selecting *TBX1* for further investigation in the revision (Pages 12-13, lines 223-231).

- It would advantageous if the authors make the genetics data available for the scientific community who would like to replicate this work.

We are willing to update summary statistics in our website (JENGER, <http://jenger.riken.jp/>) like other phenotypes after publication of our manuscript.

- Regarding the GWAS QC, cryptic relationships should also be taken into account when performing the general QC on each GWAS, not only at the final step as done in this study.

As the reviewer indicated, cryptic relationships should be taken into account in each GWAS. In fact, we applied QC criteria to each GWAS separately (and jointly to avoid cryptic relatedness found across different datasets). We have modified the text to clearly mention this point (Page 20, lines 369-371).

- It is unclear how the authors selected the causal SNP in each locus after the functional annotation. A ranking method should have been used to identify the SNP with highest functional qualities to be proposed as the one driving the association. Nowadays, there are various ranking software, like SuRFR, that can be used for this purpose.

As suggested, we have clarified how we selected the causal SNP in each locus after the functional annotation is clarified (Page 13, lines 231-237). As the reviewer indicated, there are various ranking software like SuRFR that can be used to select functional SNPs. We would like to consider the use of such software in future studies.

- The authors only discussed the biological implication of three out of 14 novel loci identified in the study. It would be important to include a summary of the known biological role for the remaining loci.

As suggested, we have included a summary of the known biological role for the remaining loci (Supplementary Table 12).

- When performing principal component analysis, the authors did not clarify the threshold used to exclude individuals.

As suggested, we have added a description to define outliers in PCA as follows. “We excluded outliers from the East Asian cluster (distance from mean of the cluster should be within 3SD).” (Page 19, lines 348-349).

- In the introduction, the prevalence for AIS is referred to the USA population, but no information is provided regarding the Japanese population.

We have provided information on the Japanese population (Page 6, lines 89-90).

- The authors miss to review one of the latest publications on AIS using exome sequencing, published in Nature Communications in 2018 (Haller *et al.*).

As suggested, we have reviewed and included the publication (Reference 32).

- The minor allele frequency threshold for imputed SNPs (0.005) is too small. It might possible be a typo, otherwise, the statistical power in this study would not be enough to identify reliable signals at this very low frequency.

We applied threshold of minor allele frequency of 0.005 after imputation. This is because this study contained subjects more than 70,000 and we have a certain power to detect signals with such low allele frequency (0.005), if variants have high effect sizes. We discussed this point in Supplementary Table 14 and Supplementary Note.

- Table 1 needs to include minor allele frequency data.

Allele frequency data are shown in Table 1 as the risk allele frequency (RAF).

- The number of SNPs genotyped and imputed used for association analysis needs to be listed. It would also be good to have information whether the SNPs listed in Table 1 were genotyped or imputed.

As suggested, we have provided information on the number of SNPs genotyped and imputed (Supplementary Table 13). Table 1 was modified to include information of being genotyped or imputed accordingly.

Reviewers' Comments:

Reviewer #1:

Remarks to the Author:

A word of thanks to the authors for the effort made to address the reviewer comments and also apologies for my belated submission of comments. In general, I am satisfied with the answers provided to the points I raised during revision. There are some remaining points I consider the authors can address better.

- I find the figure of genetic architecture (effect size vs MAF) much more informative than the Manhattan plot, but I believe there is space enough to keep both in the main manuscript.
- On a related point, with my request to compare the effect sizes of the identified variants to those efforts drawn in non-Asian populations can be discussed in the manuscript. While I agree the studies in non-Asian populations may not be powered, a detail analysis of the differences in allele frequencies across populations will be relevant and useful to guide cross-ethnic replication and complement the description of the genetic architecture. A supplementary table listing those variants in loci showing highly contrasting minor allele frequencies (greater in Asians) will allow discussing the ethnic specificity of the findings under the light of differences in minor allele frequencies.
- Finally, the table with sex stratification should contain MAF information. Also, rs73235136 is much more frequent in Asian than non-Asian populations so please discuss in that context.

Reviewer #2:

Remarks to the Author:

The new version of the manuscript submitted by Kou et al entitled "Genome-wide association study identifies 14 susceptibility loci for adolescent idiopathic scoliosis in Japanese" addressed all the concerns arisen during the reviewing process. The quality of the initially submitted paper has improved with the amendments and, therefore, I consider this work could be accepted for publication in this journal.

Answer to Reviewers' comments:

to Reviewer #1

- I find the figure of genetic architecture (effect size vs MAF) much more informative than the Manhattan plot, but I believe there is space enough to keep both in the main manuscript.

As suggested, we have included a figure of genetic architecture (effect size vs MAF) in the main manuscript (Figure 2).

- On a related point, with my request to compare the effect sizes of the identified variants to those efforts drawn in non-Asian populations can be discussed in the manuscript. While I agree the studies in non-Asian populations may not be powered, a detail analysis of the differences in allele frequencies across populations will be relevant and useful to guide cross-ethnic replication and complement the description of the genetic architecture. A supplementary table listing those variants in loci showing highly contrasting minor allele frequencies (greater in Asians) will allow discussing the ethnic specificity of the findings under the light of differences in minor allele frequencies.

As suggested, we have added a Supplementary Table of multi-ethnic MAFs for the significant loci (Supplementary Table 1). We have also compared the relationship between allele frequency and effect size, and discussed ethnic specificity (Page 8, lines 138-144 and Pages 14-15, lines 267-285).

- Finally, the table with sex stratification should contain MAF information. Also, rs73235136 is much more frequent in Asian than non-Asian populations so please discuss in that context.

As suggested, we have added the allele frequency data as the risk allele frequency (RAF) in Table 2. We also have discussed the impacts of MAF differences between JPN and EUR on rs73235136 on cross-ethnic replication (Page 15, lines 278-280).

to Reviewer #2

The new version of the manuscript submitted by Kou et al entitled “Genome-wide association study identifies 14 susceptibility loci for adolescent idiopathic scoliosis in Japanese” addressed all the concerns arisen during the reviewing process. The quality of the initially submitted paper has improved with the amendments and, therefore, I consider this work could be accepted for publication in this journal.

We thank the reviewer for the comments.